# Effect of Long-Term Use of Alcohol-Containing Handwashing Gels on the Biofilm-Forming Capacity of *Staphylococcus epidermidis*

**DOI:** 10.3390/ijerph20065037

**Published:** 2023-03-13

**Authors:** Rosa M. Lopez-Gigosos, Eloisa Mariscal-Lopez, Mario Gutierrez-Bedmar, Alberto Mariscal

**Affiliations:** 1Department of Preventive Medicine and Public Health, Malaga University, 29016 Málaga, Spain; gigosos@uma.es (R.M.L.-G.); bedmar@uma.es (M.G.-B.); 2Instituto de Investigación Biomédica de Málaga-IBIMA, 29590 Málaga, Spain; 3Hospital Costa del Sol, 29603 Marbella, Spain; 4CIBERCV Cardiovascular Diseases, Carlos III Health Institute, 28029 Madrid, Spain

**Keywords:** *Staphylococcus epidermidis*, hand disinfection, biofilm, alcohol-based gel

## Abstract

The spread of coronavirus disease 2019 (COVID-19) has promoted the use of hand sanitizers among the general population as recommended by health authorities. Alcohols, which are used in many hand sanitizers, have been shown to promotes the formation of biofilms by certain bacteria and to increase bacterial resistance to disinfection. We investigated the effect of continued use of alcohol-based gel hand sanitizer on biofilm formation by the *Staphylococcus epidermidis* resident strain isolated from the hands of health science students. Hand microbes were counted before and after handwashing, and the ability to produce biofilms was investigated. We found that 179 (84.8%) strains of *S. epidermidis* isolated from hands had the ability to form biofilm (biofilm-positive strains) in an alcohol-free culture medium. Furthermore, the presence of alcohol in the culture medium induced biofilm formation in 13 (40.6%) of the biofilm-negative strains and increased biofilm production in 111 (76.6%) strains, which were classified as low-grade biofilm-producing. Based on our findings, there is no clear evidence that the continued use of alcohol-based gels results in the selection of strains with the capacity to form biofilms. However, other disinfectant formulations that are more commonly used in clinical settings, such as alcohol-based hand-rub solutions, should be tested for their long-term effects.

## 1. Introduction

*Staphylococcus epidermidis* is the bacterial species most frequently isolated from the skin and mucous membranes of humans and other mammals [1,2]. Although *S. epidermidis* has always been considered a commensal microorganism that does not have special virulence, its role as an opportunistic pathogen gained much attention in recent years [1]. The presence of *S. epidermidis*, along with many other microorganisms (including enterobacteria and *Streptococcus* species; another coagulase-negative staphylococcus, *S. aureus*; and *Acinetobacter baumannii*), is common on healthcare providers’ hands [3]. The ubiquity and genetic characteristics of *S. epidermidis*, which make it highly resistant to harsh environmental conditions, mean it is one of the most frequent causes of nosocomial infection [1,4]. Many of these infections are related to the use of indwelling medical devices, such as peripheral or central intravenous catheters; in such cases, infection is caused by bodily penetration of the microorganism from the skin of the patient or healthcare personnel during device insertion [5,6]. Like many other human pathogenic bacteria, *S. epidermidis* can form biofilms, which are highly organized multicellular complexes embedded within a complex matrix. Biofilm formation is related to bacterial survival and resistance to antibiotics [1,7].

Hand hygiene is a fundamental aspect of reducing transmission of infections, and numerous studies show a decline in nosocomial infection rates when compliance with hand hygiene protocol is enhanced [8,9,10]. An efficient handwashing regimen depends on multiple factors, and the World Health Organization developed guidelines for the global application of clear standards to avoid patient risks related to ineffective hand hygiene [11,12]. Alcohol-based solutions have a broad spectrum of antimicrobial activity, rapid efficacy, good skin tolerance, and a clear positive effect on the rate of compliance with hand hygiene protocols; thus, such solutions are among the most widely used and recommended for hand hygiene in hospitals [13,14].

In recent years, alcohol-based hand sanitizers emerged as an important tool in the fight against severe acute respiratory syndrome coronavirus 2 (SARS-CoV-2), the virus that causes coronavirus disease 2019 (COVID-19) [15]. Washing or sanitizing hands with soap or >60% alcoholic hand sanitizer has been one of the most frequently recommended tips from the World Health Organization since COVID-19 was declared a pandemic. Alcohol-based hand sanitizers have been shown to be effective against SARS-CoV-2 and are now globally used, even in outpatient settings [15]. Although dermal contact with ethanol causes irritation and allergic conditions of the skin and eyes following prolonged exposure, several studies show that alcohol-based hand rubs are safe to use [16,17]. Since the outbreak of SARS-CoV-2, health authorities have emphasized the importance of personal hygiene in minimizing the number of infections. Increased hygiene measures have sometimes resulted in excessive handwashing and disinfection, which may affect the natural microflora and protective barrier of the skin. These measures can even have negative implications, such as toxic irritant contact dermatitis or allergic contact dermatitis [18]. Among the various formulations of alcohol-based hand sanitizers (liquids, gels, sprays, foams, and wipes), low-viscosity liquids and gels are the most common delivery modes sold to the general public because they have greater ease of handling compared to liquids [15]. Although some studies show that gels are less effective than liquids at reducing the number of microorganisms, the higher alcohol concentration in gels makes it possible to meet the European Norm (EN) test criteria (EN 1500 for hygienic hand rub and EN 12791 for surgical hand preparation) [19]. Most hand-sanitizing gels sold in Europe, which contain 80% ethanol or 75% isopropanol on a weight instead of volume basis, typically meet the above-mentioned requirements.

Microbial biofilms received special attention in recent decades [20]. The importance of biofilms is well known in the medical, industrial, and environmental contexts, mainly because of their high resistance to antibiotics and disinfectants, as well as their resistance to host immune system clearance [21]. Biofilm formation allows bacteria to survive in hostile environments, such as within the human host, and is dependent on different parameters (carbon source and concentration, pH, ionic strength, temperature, and others) [20]. The ability of *S. epidermidis* to form biofilms on surfaces is considered the main pathogenic factor of this species [22]. Biofilm production by *S. aureus* and *S. epidermidis* has been extensively studied, and the implications of a polysaccharide intercellular adhesin encoded by the ica operon, as well as protein factors such as Aap, are well known [22]. Moreover, the stress produced by some substances, such as alcohols, used for cutaneous antisepsis in clinical interventions, in addition to hand disinfection, induces the production of biofilms in numerous ica-positive strains of *S. epidermidis* [2].

Disinfectant alcohols for clinical use have proven bactericidal efficacy according to well-established standard protocols; however, not all bacteria are eliminated from the skin, and a small number may remain viable [23]. In addition, disinfected areas can be quickly recolonized by the flora of the skin areas not reached by a sufficient concentration of an alcoholic solution [2,24]. It was hypothesized that the production of biofilm by *S. epidermidis* can be either constitutive or induced by environmental factors, such as alcohol used as a skin disinfectant [2]. We investigated the effect of an alcohol-based gel used in hand disinfection on biofilm production using commensal *S. epidermidis* strains. We isolated *S. epidermidis* from the hands of volunteers before and after several washes with an alcohol-based hand-sanitizing gel, and we compared the biofilm production capacity of the isolated strains in both periods.

## 2. Materials and Methods

### 2.1. Products and Media

The hand-rub preparation used in this study was a commercial alcohol-based hand disinfectant in the form of an alcohol-based gel (HAG) containing 70% *v*/*v* ethanol (Aniosgel; Instrunet, Barcelona, Spain). Trypticase soy broth (TSB) (Oxoid, Hampshire, UK) containing a neutralizer (TSB-N), a combination of 3% Tween 80, 3% saponin, 0.1% histidine, and 0.1% cysteine, described as valid for the neutralization of alcohols [25], was used for the recovery of hand bacteria according to the procedure described in European Standard EN 1500 [26]. The neutralizing agent that is not toxic to the test organisms was used to neutralize any remaining antimicrobial activity due to disinfectant residues present in the hand swab collection fluid. Columbia Agar with Sheep Blood PLUS (Oxoid) and mannitol salt agar (MSA) (Oxoid) were used to perform plate counts of *Staphylococcus* bacteria.

### 2.2. Participants and Procedure

This study involved health science students at the University of Malaga, Spain, including nursing, occupational therapy, podiatry, and physiotherapy students, who had not been previously trained in hand hygiene. Only volunteers with intact skin and good general health were enrolled in the study. After obtaining the volunteers’ consent, we randomly selected two groups of students for inclusion in the study (each group comprised 50% male students and 50% female students). No identifying information was collected from the participants.

In the first group of students, between 9:00 and 10:00 am on 5 consecutive days (Monday–Friday), bacteria were recovered from each participant (Time 0) by rubbing the fingertips of each hand separately for 60 s in a Petri dish containing 10 mL of TSB-N. Subsequently, after 4 h (between 13:00 and 14:00 h), this group of volunteers was summoned to the laboratory (Time 1) to perform a second bacterial recovery of the hands under the same conditions as before. During the 4 h interval between Times 0 and 1, the students were instructed not to use soaps or hand sanitizers and not to modify the normal use of their hands in relation to surfaces (use of computers, mobile phones, pens, doorknobs, and other objects). This procedure was repeated for 3 consecutive weeks with the same groups of volunteers as in the first week.

The same above-described procedure was followed for the second group of students. However, during the 4 h interval between Times 0 and 1, the volunteers were instructed to perform three hygienic hand rubs with HAG, with an interval of at least 1 h between each rub, on each of the 5 days of the week. The volunteers were previously trained in the correct performance of hand rubbing. All hand rubs were performed for 30 s with 5 mL of HAG provided to each volunteer. At least 30 min elapsed between the last hand rubbing and the bacterial sampling at Time 1. As before, the volunteers were instructed not to modify their usual behavior regarding their hands and surfaces. This procedure was also repeated for 3 consecutive weeks with the same groups of volunteers as in the first week. Thus, the group that did not use HAG also did not use HAG during the other 2 weeks of the trial, and the group that used HAG continued to use the same HAG as in the first week.

From each sample obtained from the volunteers at Times 0 and 1, 1 mL of TSB-N was removed and appropriately diluted in the same broth. A total of 100 µL of each dilution (and undiluted sample) was plated onto MSA dishes and incubated aerobically at 37 °C for up to 24 h. Three MSA plates (replicates) were plated with each diluted or undiluted sample. After incubation, the colonies were enumerated, and the mean colony-forming units (CFUs) of the three replicates were expressed as log_10_. Values of 0 were reset to 1 for further calculations. Punctiform colonies grown on the MSA dishes were excluded from the bacterial count.

A collection of staphylococcus-like microorganisms was obtained from the MSA plates obtained at Times 0 and 1 from both hands of each volunteer on the first and last day of the 3-week sampling process (i.e., Monday of the first week and Friday of the third week, at Times 0 and 1 on both days). The staphylococcus-like colonies grown on the MSA plates were isolated on Columbia Agar with Sheep Blood PLUS, identified by Gram staining and catalase and coagulase activity, and then further confirmed using API Staph test strips (bioMérieux, Marcy-l’Étoile, France). At least two colonies of each morphological type grown on MSA were isolated for identification. Stock cultures of colonies identified as *Staphylococcus* were stored in 50% glycerol broth at −80 °C and revived on tryptic soy agar for 18 to 24 h at 37 °C before further use.

A biofilm test was performed with clonally unrelated isolated strains of *S. epidermidis* by determining adhesion to polystyrene microtiter plates, following a previously described method with slight modifications [20]. For the biofilm assay, TSB supplemented with 1.0% glucose (TSBG) to promote biofilm formation in vitro was used in all experiments [27]. Briefly, the wells of sterile flat-bottomed microtiter plates (Deltalab, SL, Barcelona, Spain) were filled with 90 µL of TSBG or TSBG containing either 2% ethanol (PanReac AppliChem, Darmstadt, Germany) or 2% propan-1-ol (PanReac AppliChem). Finally, 90 µL of a bacterial suspension of approximately 10^7^ CFU/mL from an overnight culture at 37 °C in TSBG were added to each well, except those used as a negative control. Negative control wells contained TSBG only. After incubation at 37 °C for 24 h, the culture medium was gently removed, and the wells were washed twice with phosphate-buffered saline (pH 7.4) to remove nonadherent bacteria. The adherent bacteria were then fixed with 180 µL of 99% methanol for 20 min, and the microtiter plates were emptied by simple flicking and left to air-dry overnight in an inverted position at room temperature. A total of 180 µL of 0.1% (*w*/*v*) crystal violet solution (Sigma-Aldrich, St. Louis, MO, USA) was added to each microtiter plate well for 15 min and left at room temperature for 30 min. The excess stain was then discarded, and the plate was washed twice with running water to remove residual dye. After drying for 30 min at room temperature, 180 µL of absolute ethanol were added to elute the stain-adherent biofilm, and the plate was left at room temperature for 30 min without shaking. The optical density (OD) at 570 nm (OD570) of each stained well was measured using a microplate reader (RT-6500; Rayto, Shenzhen, China). Following previously established criteria [24], the strains were classified according to biofilm formation into biofilm-negative strains (OD570 < 0.1), low-grade biofilm-producing strains (0.1 ≤ OD570 < 1.0), and strongly biofilm-producing strains (OD570 ≥ 1.0). *Staphylococcus epidermidis* ATCC 12228 was used as the control biofilm-negative strain. Increased biofilm formation due to ethanol or propan-1-ol was defined as at least the doubling of the OD of strains in the absence of alcohol. All biofilm formation experiments were performed in triplicate and performed twice on different days. The OD values were then averaged, and the standard deviation was calculated.

### 2.3. Statistical Analysis

Nonparametric methods (the Friedman test or Kruskal–Wallis test) were used to test for differences between the groups with or without HAG handwashing at different test times. The differences among the OD values of the samples and controls of the biofilms were assessed using analysis of variance and a one-tailed test assuming equal variance via the Microsoft Excel data analysis tool. Statistical significance was defined as *p* < 0.05.

## 3. Results

The results from the 20 volunteers (aged 19–21 years) were analyzed. In the volunteers who did not use HAG (non-HAG group) between the first (Time 0) and second recovery of samples, the Friedman test showed a significant increase in the mean number of microorganisms isolated from the hands at Time 1 (5.62 ± 5.53 and 6.24 ± 6.43 log_10_ CFU of 3 weeks for Times 0 and 1, respectively; *p* = 0.0133). In the volunteers who used HAG three times daily between the two sample collections (Times 0 and 1), a significant reduction was observed in the mean CFU (5.93 ± 6.16 and 4.51 ± 4.80 log_10_ CFU of 3 weeks for Times 0 and 1, respectively; *p* = 0.00006). Treatment with HAG caused a mean bacterial reduction of 1.42 log_10_ CFU, although a large amount of variation was observed in the bacterial counts obtained between the participants, ranging from 1.00 log_10_ CFU (0.00 CFU per hand) to 5.30 log_10_ CFU at Time 1. Figure 1 shows the mean bacterial counts (presented as mean log_10_ of CFU) of the samples recovered from Monday to Friday of each week, obtained at Times 0 and 1, for the total number of participants during the 3-week study period. The Friedman test showed no significant difference between the right- and left-hand bacterial counts for the two groups (*p* = 0.467 and *p* = 0.193 for non-HAG and HAG groups, respectively), nor between the daily (15 days) bacterial counts obtained from the 20 volunteers at Time 0 (*p* = 0.115, non-HAG and HAG groups). Therefore, Figure 1 shows the mean counts from both hands obtained each week for each group of participants.

In total, 516 isolates with suspicious staphylococcal morphology were obtained from the hands of the volunteers on the first and last days of the assay, as described in Section 2. In general, one to five colonies with different aspects were investigated on each plate; however, in the HAG group at Time 1, no colony growth was observed on the MSA plates of any of the volunteers. Fifty-four isolates were later discarded because the Gram staining or the biochemical screening tests were not compatible with staphylococci. Microorganisms suspected of having been repeated because they were isolated from the same count plate and because they had the same morphological and biochemical characteristics were excluded from further study. Finally, 196 staphylococcus-like microorganisms at Time 0 and 160 at Time 1 (Table 1) were identified. *Staphylococcus epidermidis* was the most frequent species, with 211 strains isolated. Among the other species identified, the most frequent were *S. capitis* (*n* = 35), *S. aureus* (*n* = 32), *S. warneri* (*n* = 29), and *S. haemolyticus* (*n* = 18). The Kruskal–Wallis test showed no significant differences in the distribution of identified staphylococci species between the test groups (non-HAG vs. HAG, *p* = 0.531).

*Staphylococcus epidermidis* biofilm formation was analyzed in TSBG, TSBG supplemented with ethanol, and TSBG supplemented with propan-1-ol at subinhibitory concentrations. Of the 211 microorganisms studied, 179 (84.8%) were classified as biofilm-producing in TSBG, of which 33 (18.4%) were classified as highly biofilm-producing, and 146 (81.6%) were classified as low-grade biofilm-producing. Ethanol and propan-1-ol induced biofilm formation in 13 and 2 TSBG biofilm-negative strains and increased biofilm production in 111 and 75 TSBG low-grade biofilm-producing strains, respectively. In all strains for which biofilm production was induced or increased by propan-1-ol (77 strains), ethanol produced the same effect, inducing or increasing biofilm production. However, in 42 of the 119 strains with biofilm production induced or increased by ethanol, propan-1-ol did not produce the same effect. Overall, handwashing resulted in no significant differences in the proportion of strains according to the ability of ethanol to induce biofilm production in biofilm-negative strains or increase biofilm production in low-grade biofilm-producing strains obtained at different stages of the study (Kruskal–Wallis test, *p* = 0.967 and *p* = 0.739 for the non-HAG and HAG groups, respectively) (Figure 2). However, as mentioned above, the ability of these alcohols to increase biofilm production in the 33 highly biofilm-producing strains could not be detected because of the technical limitations of the photometer. The proportion of highly biofilm-producing strains at Time 1 on the last day was higher than at the other times tested, although the differences were not statistically significant.

## 4. Discussion

Frequent and careful handwashing by health personnel is the most important measure in the prevention of hospital infections, and its ability to reduce the chances of contracting COVID-19 was recently suggested [28]. Backed by the World Health Organization and most health authorities, hand hygiene has played a prominent role in COVID-19 preventive strategies. Dispensers with hand sanitizers have been placed in most public spaces, and numerous campaigns have been promoted to emphasize the importance of their use to prevent the spread of SARS-CoV-2 [29,30,31]. Regular handwashing with soap or disinfectant gels containing alcohol has been a widespread practice among health workers and the general population, especially since the early days of the pandemic [30]. However, ethanol and other alcohols used in many hand sanitizers can reportedly increase biofilm formation in microorganisms that, like *S. epidermidis*, may be relevant to the development of nosocomial infections [24]. In this study, we investigated biofilm production in 211 strains of *S. epidermidis* obtained in a trial involving health science students in which handwashing conditions comparable to a real situation of frequent handwashing were simulated. Biofilm production was studied in strains obtained from two groups of students over a period of three consecutive weeks; one group used a handwashing gel containing ethanol, whereas the other group did not.

During the study, more than five *Staphylococcus* species were identified from the participants’ hands, and as expected, *S. epidermidis* was the most dominant species [3]. The estimated number of microorganisms calculated in the counts from the hands and the specific species identified in our study correspond to a representative sample of microorganisms that are most frequently isolated on the hands of health personnel [3,32]. As in other studies [3,33], we found that both hands of all participants were contaminated in the counts made prior to the use of disinfectant gel, which was expected in this study because the participants had previously made contact with surfaces that had not been disinfected (furniture, hands, their own skin, and handrails). The counts at Time 0 were very similar between the volunteers who did and did not use disinfectant gel (5.93 ± 6.16 and 5.62 ± 5.53 log_10_ CFU, respectively). In contrast, the count was significantly higher at Time 1 than at Time 0 in the non-HAG group, but it was significantly lower at Time 1 than at Time 0 in the HAG group, clearly showing the antimicrobial effect of the disinfectant gel. After rubbing hands with HAG according to the experimental protocol (three washes in 4 h per day), the average bacterial reduction obtained (1.42 log_10_ CFU) was lower than that recommended for similar gels containing ethanol adapted to the EN 1500 standard [34]. These results were expected because, after the third wash (every day) with HAG, the volunteers continued to use their hands as usual with respect to surfaces for at least 30 min. However, the concentration of microorganisms in this group of participants (HAG group) at Time 1 was highly variable, with counts ranging from 0 to more than 10^5^ CFU per hand. As indicated above, because there was no statistically significant difference, the data reflected in Figure 1 show the mean counts in both hands obtained in the 15 days for all participants. Our trial was designed to approximate a real situation to the greatest extent possible; therefore, it is expected that despite the use of HAG, cross-contamination of the hands may have occurred from other parts of the body or from everyday surfaces with which the hands came into contact during the 30 min between the last hand rubbing and sampling. No bacteria or other microorganisms with resistance to ethanol have been reported to date, and the efficacy of alcoholic solutions according to pre-established criteria has been widely demonstrated [35]. Thus, most of the strains detected in the HAG group at Time 1 were likely a result of this cross-contamination related to the habits of each participant. This can be deduced from the fact that in the counts at Time 1 in this group, 56% of the participants had a count of 0 CFUs on the MSA dishes according to the sampling method performed.

In our study, we focused on examining biofilm production under standard conditions, including NaCl and glucose at concentrations previously known to enhance biofilm formation [27], and in the presence of ethanol and propan-1-ol. The concentrations of ethanol and propan-1-ol used in our trial were lower than those used in alcohol-based sanitizers recommended for sanitizing health workers’ hands. Although these disinfectant solutions are known for their bactericidal efficacy in clinical use, it is assumed that a small number of bacteria could survive after skin disinfection [2], facilitating recolonization of the skin by flora near the disinfected area, other inert surfaces, or contaminated skin. When the ability of ethanol and propan-1-ol to induce biofilm production in TSBG-negative biofilm-producing strains was investigated, 19 strains remained biofilm-negative under all conditions. However, of the 13 ethanol-induced TSBG-negative biofilm-producing strains, 2 strains were also propan-1-ol-induced, but no strain was induced only by propan-1-ol. This difference between propan-1-ol and ethanol with respect to its effect on biofilm production was also observed in the 146 biofilm-positive strains in which it could be studied (low-grade biofilm-producing strains); ethanol and propan-1-ol increased biofilm production in 111 (76.0%) and 75 (51.4%) strains, respectively. As with TSBG-negative biofilm-producing strains, low-grade biofilm-producing strains that increased biofilm production upon addition of propan-1-ol also increased biofilm production when ethanol was added; however, in 36 strains with increased biofilm production in the presence of ethanol, propan-1-ol did not produce a similar effect. The proportion of biofilm-positive strains inducible by at least one of the alcohols used in our study was within the ranges observed in other studies (31–82%), although the proportion of initially biofilm-negative strains inducible by the action of at least one of the alcohols (40.6%) was higher than the proportions observed in these studies (4–11%) [2,24]. However, in these studies, neither the origin of the strains (usually from clinical samples) nor the alcohol concentration used in the biofilm induction studies was the same; therefore, the results are not entirely comparable.

It has been argued that the induction of biofilm production on the skin around a disinfection site, where disinfectant concentrations are lower, could be of concern because it may promote healthcare-associated infections [2]. In our study, phenotypic analysis of biofilm expression revealed that the capacity to produce biofilm among the *S. epidermidis* strains studied was high (69.2%), and this percentage rose to 75.3% (including biofilm-negative strains) in the presence of low ethanol concentrations. Furthermore, more than two-thirds of the strains showed increased biofilm production in the presence of low alcohol concentrations.

It has been speculated that the continued use of disinfectants, such as alcohol, which induce biofilm formation, could exert positive selective pressure on biofilm-positive microorganisms on the skin and favor the development of infections in healthcare settings [24]. However, after the use of alcohol-based disinfectant solutions for at least 3 weeks under the experimental conditions tested in our study, we did not observe a significant increase in biofilm-producing strains on the hands.

The findings of this study are limited by the fact that the microorganisms analyzed constituted only a small sample of microorganisms present on the skin and were limited to *S. epidermidis* as a single species. There was no significant difference in the number of microorganisms isolated during the first sampling (Time 0) on the hands of participants between the non-HAG and HAG groups throughout the 3-week study period, indicating that the population of microorganisms remained stable even in the group that performed continuous handwashing with HAG. However, in the HAG group, although the reduction in microorganisms from Time 0 to Time 1 was statistically significant and although no growth of microorganisms such as *Staphylococcus* was observed on many hands, in some students, the number of microorganisms was restored to levels close to those at Time 0 only 30 min after the last washing with HAG. This suggests rapid colonization from different microbiological environments in the work environment, including microorganisms from other parts of the volunteers’ skin through hand contact. Handwashing with HAG applied correctly removes most of the microorganisms present on this part of the skin, but after a relatively short time, the hands are again occupied by microorganisms from surfaces and other parts of the skin. This might explain why no differences were observed between the non-HAG and HAG groups in terms of the number of biofilm-producing microorganisms or in the number of microorganisms in which biofilm production was increased in the presence of alcohol in the hand sanitizer. However, this study was conducted with alcohol-containing gels; liquid hand rubs commonly used in hospital practice, which showed higher disinfectant efficacy in some studies [36], could exert higher selective pressure and favor biofilm-forming strains.

## 5. Conclusions

The results of this study not only suggest the importance of proper hand hygiene before and after contact with any surface, especially in healthcare settings, but also reveal the high number of biofilm-producing isolates that display biofilm induction by alcohol. Further studies involving the microorganisms most known to be responsible for hospital infections and studies that reproduce real-life scenarios should be conducted in hospital settings where liquid alcoholic rub-in hand disinfectant is routinely used.

## Figures and Tables

**Figure 1 ijerph-20-05037-f001:**
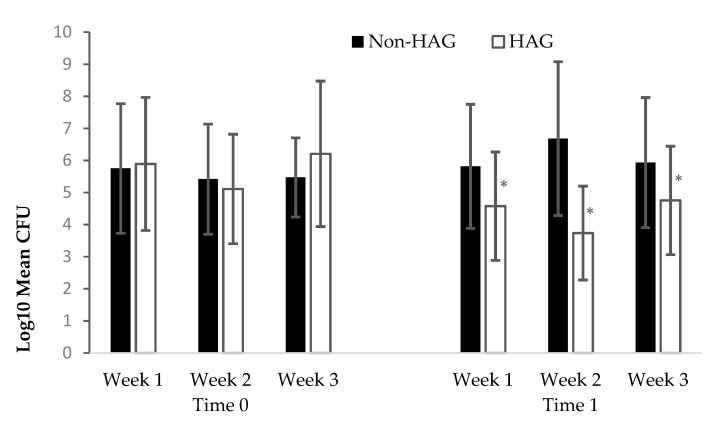
Recovery of bacteria from the hands of volunteers according to the use of alcohol-based gel (HAG). Values are expressed as log_10_ mean CFU of 5 days a week for 3 weeks for left and right hands. Error bars indicate 95% confidence interval. Non-HAG: volunteers who did not use alcohol-based gel; HAG: volunteers who used alcohol-based gel. A statistically significant decrease in the counts (as log_10_ CFU) obtained each week between Time 1 and Time 0 according to Friedman’s test is denoted by an asterisk (*p* < 0.05).

**Figure 2 ijerph-20-05037-f002:**
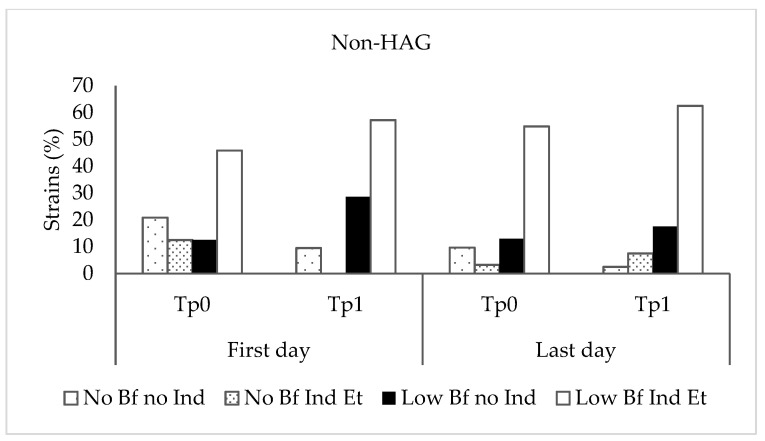
Proportion of strains according to the ability of ethanol to induce or increase biofilm production at different stages of the study. Non-HAG: volunteers who did not use alcohol-based gel; HAG: volunteers who used alcohol-based gel. No BF no Ind: biofilm-negative strains, non-ethanol-inducible; No BF Ind Et: biofilm-negative strains, ethanol-inducible; Low BF no Ind: low-grade biofilm-producing strains, not inducible by ethanol; Low BF Ind Et: low-grade biofilm-producing strains, ethanol-inducible.

**Table 1 ijerph-20-05037-t001:** Most frequently isolated microorganisms on MSA plates at different stages of the study.

	*S. epidermidis*	*S. capitis*	*S. aureus*	*S. warnerii*	*S. haemolyticus*	Other
Non-HAG						
Time 0	55	14	9	3	2	12
Time 1	61	5	11	13	8	5
HAG						
Time 0	59	14	9	7	9	3
Time 1	36	2	4	6	0	9

## Data Availability

Not applicable.

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
