# Peer review of "Effect of Long-Term Use of Alcohol-Containing Handwashing Gels on the Biofilm-Forming Capacity of Staphylococcus epidermidis"

_ijerph, 2023, doi:10.3390/ijerph20065037_

Round 1

Reviewer 1 Report

Lopez-Gigosos et. al., have done a very interesting work. I would suggest few things if you can address,

1. Authors used two groups for this study, HAG, and non-HAG group. Please clarify the 2 subgroups mentioned in line 121.

2. Please provide a table with volunteer’s number in each group with age and sex.

3. In figure 1 authors did not mention the name of white bar.

4. I would suggest seeing the effect of Ethanol/propanol at different subinhibitory/suboptimum concentrations on biofilm formation.

5. It will strengthen the work if you can show images of biofilm treated with or without ethanol/propanol. It would be great if you can quantify the total biofilm protein concentration.

Author Response

We would like to thank the reviews provided by the editors and each of the reviewers on our manuscript Re: ijerph-2248979, entitled “Effect of long-term use of alcohol-containing handwashing gels on the biofilm-forming capacity of Staphylococcus epidermidis.”. The comments are encouraging, and we have followed then to make appropriate changes in the paper. We are sure that these changes will improve the quality and understanding of our work. Please see below, a point-by-point detailed response to the comments.

Response to reviewers’ comments

Comment Reviewer 1

General comments:

Lopez-Gigosos et. al., have done a very interesting work. 1 would suggest few things if you can address,

Authors’ Response

Thank you very much for your kind comments and suggestions, which will undoubtedly improve the understanding of our article.

Specific comments:

  1. Authors used two groups for this study, HAG, and non- V HAG group. Please clarify the 2 subgroups mentioned in line 129-132.

Authors’ Response

Thanks for the observation. Indeed, there is an error in the wording of the paragraph. In the new version of the manuscript, we have corrected this error. Please find the changes made to the paragraph beginning on line 121.

  1. Please provide a table with volunteer's number in each group with age and sex.

Authors’ Response

In our study we have used two very homogeneous groups of students, 50% of each sex (male/female) and a very narrow age range (19 to 21 years old), so we believe that a table could be somewhat excessive as information when not affect the objectives and results of the study. No other personal information was obtained from the volunteers. However, in the new version we have incorporated the age range of the students (first paragraph of Results).

  1. In figure 1 authors did not mention the name of white bar.

Authors’ Response:

Thank you for your comment. Certainly, the caption had been cropped by changing the font sizes in the figure. In the new version this problem has been corrected.

  1. I would suggest seeing the effect of Ethanol/propanol at different subinhibitory/suboptimum concentrations on biofilm formation.

Authors’ Response:

The effect of different sub-inhibitory concentrations of ethanol and propanol on biofilm formation was previously analyzed by another author. The results did not show remarkable differences when the concentration of these alcohols was close to the inhibitory concentration. For this reason, in our study, only the sub-inhibitory concentration closest to the inhibitory concentration has been studied.

  1. It will strengthen the work if you can show images of biofilm treated with or without ethanol/propanol. lt would be great if you can quantify the total biofilm protein concentration.

Authors’ Response:

We are sorry, but in our study, we only determined the optical density of the biofilm by spectrophotometry, but no pictures were taken of any of the experiments. On the other hand, biofilm density was only estimated from the crystal violet staining technique, probably the most used technique, when it is not a study of the intrinsic characteristics of the biofilm. In other types of studies, e.g. on biofilm quality, it would have been interesting to include them.

We hope that the reviewers will find the changes made in this new version satisfactory, and we remain at your disposal for any other suggestions that contribute to a better precision of our manuscript.

Sincerely,

Alberto Mariscal (as a representative of all authors)

Reviewer 2 Report

Dear Authors,

Thanks for the study. I have reviewed the paper thoroughly. I realize that there are some fundamental errors in the manuscript. Either I am not able to understand or the authors could explain well. Because of that sentences meanings are getting diverted and not understandable. 

Please revise the manuscript taking help from a colleague who is proficient in English and familiar with the subject matter, who can review your manuscript, or contact a professional editing service to review your manuscript. It was very hard to read the manuscript at various places, I started to fix some mistakes, but then I gave up as there is a problem in many sentences; either the sentence is too short, the words used in the sentence are not the best choice or not used normally in that context, grammatically wrong or the author did not deliver the message (meaningless sentences), huge amount of typo errors, capital and Italics. Please revise the manuscript thoroughly for English language issues.

For example: Staphylococcus epidermidis in the title itself is not written in Italics. 

Figure 1 legend: hidroalcoholic gel spelling

Nonadherent must be written as non-adherent.

Log10 must be written as log10 with subscript.

Please check thoroughly throughout line by line. Many mistakes like this.

Regarding major comments, in abstract authors mentioned 

“The 84.8% of strains isolated had the ability to form biofilm. The presence of alcohol in the culture medium induced biofilm formation in 40.6% of biofilm-negative strains”

I do not understand this. Around 84.8% strains are biofilm producer. Then how alcohol has induced 40.6% negative biofilm producers. Negative biofilm producers supposed to be 15.2% only. Please justify or clarify in the manuscript. Or you are trying to say 40.6% from 15.2%.

Authors have used rub containing 70% v/v ethanol for this study but neutralize the alcohol efficiency, If I am not wrong this is what authors meant? If you have neutralized the alcohol efficiency, then what is the significance of the study? What are you trying to portray from the results. The actual use of alcohol gels is gone, then what are you testing? Please explain.

Please explain and revise this sentence:

“90 μL of TSBG (control) or TSBG supplemented with either 2% ethanol or 2% propan-1-ol”

I do not understand here. Which one is control? 90 μL of TSBG or TSBG supplemented with either 2% ethanol or 2% propan-1-ol or both? Something is not clear here. Using “OR” is not a write choice. Its not portraying what you want to portray. What is the difference between two controls?

Why you used 2% when you used 70% ethanol containing gel?

Staphylococcus epidermidis ATCC 12228 was used as the control biofilm-negative strain” Are you sure? Did you use this as negative control or positive control? Why there is no positive control in the study?

Please explain the statistical analysis in the bar graphs. There is no statistical outputs in the figure legends and bars. What is Tp in the figures. Please elaborate in the legend for better understanding. Please make bar graphs colorful to differentiate between the bars easily.

I need to understand these fundamental issues before I proceed with another round of thorough revision. Please explain as easily as possible and similarly revise the language in the paper for a good read.

Author Response

We would like to thank the reviews provided by the editors and each of the reviewers on our manuscript Re: ijerph-2248979, entitled “Effect of long-term use of alcohol-containing handwashing gels on the biofilm-forming capacity of Staphylococcus epidermidis.”. The comments are encouraging, and we have followed then to make appropriate changes in the paper. We are sure that these changes will improve the quality and understanding of our work. Please see below, a point-by-point detailed response to the comments.

Response to reviewers’ comments

Comment Reviewer 2

 Comments to the Author

Please revise the manuscript taking help from a colleague who is proficient in English and familiar with the…

Authors’ Response

Thanks for your comments. The new manuscript has been revised by MDPI English Editing), a company we have worked with on several previous occasions to our complete satisfaction. We assume that many of the errors detected in the manuscript (words not in italics, spelling of words or use of inappropriate words such as nonadherent, hydroalcoholic, Log10, etc.) have occurred while formatting the article to adapt it to the template suggested by IJERPH. Our apologies for this. In the new document we have prepared we hope that these errors have been corrected.

1. Specific major comments:

Regarding major comments, in abstract authors mentioned “The 84,8% of strains isolated had the ability…” … Please justify or clarify in the manuscript…

Authors’ Response

Thank you for your comment. It is true that the paragraph referred to may be misleading. In the new document we have modified this paragraph while indicating the number of strains and the corresponding percentages, which we hope will make it much more understandable (line 15-19).

2. Specific major comments:

Authors have used rub containing 70% v/v ethanol for this study but neutralize the alcohol efficiency, If I am not wrong this is what authors meant?...

Authors’ Response

We have used the neutralizing agent to neutralize any residual antimicrobial activity that may remain on the hands after the use of a disinfectant. This is the usual procedure when the aim, as in our case, is to count and study the microorganisms that may have survived the use of the disinfectant for various reasons (see introduction and discussion paragraphs, lines 95-101, and 355-361). For the sampling of microorganisms from hands we have done it according to the procedure described in European Standard EN 1500 (EN 1500, 2013) for the assessment of the efficacy of hand disinfectants.

3. Specific major comments:

Please explain and revise this sentence:

"90 µL of TSBG (control) or TSBG supplemented with either 2% ethanol or 2% propan-1-ol" I do not understand here…

Authors’ Response

Thank you again for your comment. We agree that the paragraph referred to was not sufficiently clear. In our study, we have determined and compared the ability of each of the strains to produce biofilm in three different media: TSBG, TSBG+ethanol, and TSBG+propanol. As a negative control, wells containing TSBG alone were used, because the addition of alcohol at that concentration did not change the measurement in the spectrophotometer. In the new version of our paper, we have modified the paragraph beginning on line 173, which we hope will improve the comprehension of the text.

4. Specific major comments:

Why you used 2% when you used 70% ethanol containing gel?

Authors’ Response

The commercially available hand sanitiser we use in our study, which meets European standards, is 70%. For the biofilm-forming capacity test, however, we used a sub-inhibitory alcohol concentration (2%). We aim to determine whether the ability of the strains to produce or induce biofilm is somehow selected and is more frequent due to the continuous use of hand washing with an alcohol-containing disinfectant. As mentioned above, not all microorganisms are killed in the hand disinfection process for various reasons. In the in vitro test for biofilm production, we can only use sub-inhibitory concentrations, because with higher concentrations we would not obtain any growth and it would be impossible to determine whether these surviving bacteria have a higher capacity to produce biofilm.

5. Specific major comments:

"Staphylococcus epidermidis ATCC 12228 was used as the control biofilm-negative strain" Are you sure? Did you use this as negative control or positive control? Why there is no positive control in the study?

Authors’ Response

Staphylococcus epidermidis ATCC 12228 is a strain widely used in studies of disinfectants and antibiotics, which does not adhere to the walls of microtiter plates and does not form a biofilm. For these reasons, as in our study, it is a strain widely used as a negative control. On the other hand, the biofilm determination technique used is based on the ability of the strains to adhere to materials (due to the polysaccharides of their cell wall) such as microtiter plates, producing biofilm. As is known, the biofilm determination technique used is a colorimetric method based on crystal violet. Bacteria that could adhere (forming biofilm) to materials retain the dye in their polysaccharide layer. However, the strains that do not form a biofilm remain in the supernatant and are washed away during the assay, with the wells appearing practically colorless. The presence of color in the wells, with statistically significant differences with respect to the strains that do not produce biofilm, allows us to know if a strain has adhered (forming biofilm) and in what quantity it produces it.

6. Specific major comments:

Please explain the statistical analysis in the bar graphs.

There is no statistical outputs in the figure legends…

Authors’ Response

Thank you for your comment. For a better understanding, in the new version we have replaced the paragraph related to the statistical notations of the figure.

We look forward to hearing from you in due time regarding our submission and to respond to any further question and comments you may have.

Sincerely,

Alberto Mariscal (as a representative of all authors)

Round 2

Reviewer 1 Report

Thanks for addressing most of the questions but would suggest to take pictures of biofilms and quantifying total protein in future while doing anti-biofilm study. 

I would be happy to recommend the work for publication.

Author Response

Again, thank you very much for your kind comments. We will take your comments into consideration for future works. Certainly, it is always appropriate to take photos of the experiences.  Our group has published several works with biofilms of different organisms, but we have not included photographs in any of them. All these works have been carried out on dozens of microorganisms which, with their corresponding replicates, amount to more than a hundred microtiter plates. For this reason, we have always thought that photographs add little information about the results and could lead to an overload of material in publications. Another case would be if we were studying some special feature of biofilm formation, in which case it would be appropriate to publish the photograph.  In any case, we note your observation for future work. Thank you again for your efforts in revising the manuscript, and we remain at your disposal for any comments you may have.

Reviewer 2 Report

Manuscript is significantly improved by the authors. However, I am not convinced regarding the use of Staphylococcus epidermidis ATCC 12228 as negative control. I read many manuscripts and found out that various previous works on Staphylococcus epidermidis ATCC 12228 has shown mixed results regarding attachment to the microtitre plate surface. It has never been 100% valid. I suggest authors to provide the image of microtitre plate with and without Staphylococcus epidermidis ATCC 12228 and add in supplementary file to confirm the results and to provide readers with the relevant information.

Author Response

Thank you very much for your comments and effort in reviewing our manuscript.

Regarding your comments on the strain used as a negative control, it is true that this strain, like any other strain, whether biofilm-producing or not, can in certain cases give results that can be confusing to interpret. The causes can be very variable, from the use of reagents or microtiter plates of different brands to small alterations (difficult to detect) in the performance of the test (small bubbles, variations in the temperature of the different solutions used, slight variations in the sizes and age of the inocula, etc.). In general, protocols are usually very strict to avoid these differences, but in any case, following the references of the original method, for each assay, several measurements are made for each condition (on the same plate) and performed on at least three occasions (the three replicates indicated in the article). For each assay, the outlier values were identified by the Z-score method, and non-outliers data were used to calculate average values. A strain was considered to form biofilm when the ABS values were three times the standard deviation of the mean absorbance of negative controls. These issues are included in the original methodology and therefore we have not comment them in our manuscript as we have not changed anything in this respect.

Regarding your suggestion to provide a picture of the microtiter plate, as we commented to reviewer #1, unfortunately we have not taken pictures of the microtiter plates and all data has been based on ABS readings. This work, as well as others published by our group on biofilms, has been carried out on dozens of microorganisms which, with their corresponding replicates, add up to more than a hundred microtiter plates. For this reason, as I indicated to Reviewer #1, we have always thought that a photograph provides little information about the results, unless we are studying some special feature of biofilm formation, in which case it would be appropriate to publish the photograph. On the other hand, to add a photograph to the paper, we would have to carry out a new experiment, which would require at least two weeks (revitalization of strains, cultivation and standardization of reagents and culture media, calibration of equipment). This would require an extension of the deadline given by the editor to revise the article. In addition, it would be a photograph taken specifically outside the work developed in the article.

We naturally look forward to your suggestions. Thank you very much for your comments which have undoubtedly contributed to a better understanding of our article.

Best regards